# Convivial Connection between Staff and Customer Is Key to Maximising Profitable Experiences: An Australian Cellar Door Perspective

**DOI:** 10.3390/foods11193112

**Published:** 2022-10-06

**Authors:** Genevieve d’Ament, Tahmid Nayeem, Anthony J. Saliba

**Affiliations:** 1School of Psychology, Faculty of Business, Justice and Behavioural Sciences, Charles Sturt University, Bathurst, NSW 2795, Australia; 2School of Business, Faculty of Business, Justice and Behavioural Sciences, Charles Sturt University, Wagga Wagga, NSW 2650, Australia

**Keywords:** staffing, human resources, cellar door, co-created experience, cellar door experience, brand attachment, constructivist grounded theory, cognitive dissonance, profitability, winescape

## Abstract

Cellar doors provide retail sales for wineries, providing higher returns than wholesale to domestic and export markets. Customer-based research has established enjoyable cellar door experiences are essential to building brand attachment, creating enduring customers, and increasing on-site and post-visit sales. However, customers co-create cellar door experiences with staff, as each approach the experience with unique realities guiding their expectations. Scarce literature includes experiences from staff perspectives. Constructivist grounded theory and adopting Charmaz’s approach to analysis were used to explore data from semi-structured, in-depth interviews with 33 wine consumers ranging in wine involvement, from wine novices to highly involved enthusiasts and winemakers. Twenty-two of the consumers were cellar door staff with experiences ranging from a few months to owners of long-established family wineries. Cognitive dissonance theory helped us to understand how tensions may arise for individuals approaching each experience and where to avert perceived risks. Findings show convivial connection respecting all actors creates enjoyable experiences. The warmth of greeting, further strengthened by staff–customer rapport, developed via knowledgeable conversation throughout the experience, increases brand attachment. Co-created connection becomes the conduit through which positive experiences are created and where brand attachment is forged. A framework based on emerging categories guides professional development models and human resources strategies for wineries, thereby maximizing profitability through cellar door sales.

## 1. Introduction

Wine Australia has invested in research to ensure exceptional Australian wine quality. Great wine has perhaps become an assumption leading winery management to believe gimmicks and non-wine related products are needed to engage customers. However, that should not mean the wine is forgotten or removed from the centrepiece of the experience. Far from it, the cellar door experience (CDE) should be all about wine. However, to dispel the dissonance associated with CDEs, the experience co-created in the cellar door must be at least as good as the wine being offered.

The CDE has been well researched over the last twenty years since industry awareness was raised by O’Neill and Charters [1] and Alant and Bruwer [2]. The CD provides for direct-to-customer sales on-site, and the opportunity to develop a relationship to continue direct-to-customer sales post-visit, further adding to CD profitability, and subsequent research has proven the importance of the CDE to a winery business. Customers have been surveyed to determine the importance of a positive CDE and the influence of such on customer loyalty [3], brand awareness [4], re-visitation [5], recommendation [6], and profitability [7]. The ambience of the built environment showcasing sweeping vistas over surrounding vineyards, collectively known as the winescape, in which the CDE occurs, has drawn numerous studies promoting its role in motivating visitors to visit wine regions [8,9,10,11].

Customer behaviour research has found product exposure (e.g., bottle shop purchases) builds brand awareness, but experiences (i.e., co-creating a cellar door experience) lead to brand attachment, developing customer loyalty, thereby increasing re-visitation, post visit sales, and word-of-mouth recommendations [12]. However, customers are only part of the CDE equation. The CDE is a dyadic relationship between the cellar door (CD) staff member and at least one customer. Together they co-create the experience.

CDEs are considered multi-sensory, hedonic, and pleasurable pursuits [13], which fit under the construct of wine tourism, and therefore, much of the CDE research has been conducted within the tourism discipline [14]. Tourism research design and methodology has predominantly followed case study, survey, and analysis of online reviews (e.g., TripAdvisor). Studies not only excluded the staff perspective, but have examined CDEs through a tourism lens, and therefore include within their data collection associated activities, such as winery tours, restaurants and cafes, hotel accommodation, wine trails, and wine festivals.

As mentioned, scarce research has included the role of staff in any experience or studied an experience from a staff perspective. Carvalho et al. [15] focused on the tourist in their study of experience, inferring the only human element of the experience was the tourist who interacted with an experience (e.g., winery tour). Zatori et al. [16] found the interaction between staff and customer to be a greater influence on memorability and authenticity in sight-seeing tours than the physical environment. However, again their methodology developed a survey for customers without including the perspective of the tour guides. Additionally, the methodology used in tourism and experience research often includes focus groups to develop surveys, which have been found to be susceptible to a non-response bias [17].

Wine industry staff-focused research has established the demographics of those employed in cellar doors and their desired attributes [18] but, to our knowledge, research is yet to explore the dyadic relationship of the co-created CDE. The current study approaches the actual CDE without the inclusion of the winescape or tourism trappings. It explores both staff and customer perspectives, or the co-created CDE, focusing on the experience, which provides winery businesses the opportunity to maximize winery sales, brand loyalty, re-visitation, and word-of-mouth recommendation.

To understand the co-created CDE, our research makes an in-depth exploration of staff and customer recollections and interpretations of their CDEs. The current study seeks to determine how staff and customers co-create memorable experiences, with the following questions considered and analyzed:

How do staff establish a unique relationship with customers within a 30–60-min tasting experience?

How important is the staff’s role in developing the relationship to transform cellar door visitors into customers?

The following paper contains a brief overview of the current literature, an outline of the study’s methods and data analysis. The final sections present the findings, discussions, limitations, future research, and conclusion.

### 1.1. Literature Review

#### 1.1.1. Importance of Cellar Door

Wine is a unique sales product. In addition to bottle shop and restaurant settings, it can be tasted, with or without being purchased, in the CD, on the site where it was produced. The CD is the winery business’ retail outlet with CD sales providing greater profit margins than wholesale supply to domestic and export markets [19]. Motivations for visiting CDs to engage in tasting a winery’s product are not limited to the hedonic and pleasurable nature of a CDE [13]. Previous experience with, or knowledge of, a winery or winery’s product has been established as a motivator for travelling to that winery’s cellar door to experience their product, where their product is made [20]. In Australia, not all wine is grown and produced on the same site as the CDs. Broadly, there are three components of wine production: the vineyard, where the grapes are grown; the winery building, where the wine is made; and the CD, where customers taste the wine and make purchases. Wine businesses can: grow all of their fruit, but transport it to an off-site winery to make their wine; grow some of their fruit (sometimes only an acre under vine) and buy in fruit from growers to make wine in their winery on-site and sell it at their CD; or, as assumed by most wine tourists [21], grow their fruit in the vineyards surrounding their CD and make the wine in the winery on the same property. Therefore, a CDE is much more than a retail outlet for direct-to-customer sales.

#### 1.1.2. Perceived Risks of Wine Purchases

One reason for engaging in and purchasing wine during the CDE is to reduce the risk associated with purchasing an unknown product. Johnson and Bruwer [22] evaluated risk-reduction strategies employed by Australian wine customers when purchasing wine in a retail setting (i.e., franchise bottle shops such as Dan Murphy’s, or independent liquor stores with catchy names such as Cork & Glass). Their categories of perceived risk included functional, social, financial, and physical. Their suggested risk-reduction strategies, which could be employed by customers included information seeking, brand loyalty, store image, well-known brand, price, and reassurance. Johnson and Bruwer’s [22] perceived risks and risk-reduction strategies can be assimilated to the CDE (see Table 1). In their study, participants employed risk-reduction strategies when purchasing wine. As the bottle price increased there was a preference for trying the wine before purchase, except for the novice wine drinker, who would still prefer to stay with a known brand. Finally, the ability of retailers to recognize and reduce associated risks provided them a competitive advantage (e.g., providing tastings of a higher priced bottle of wine increased purchases of the higher priced wines). Duarte Alonso et al. [23] also recognized the need to minimize the intimidation that wine, or those presenting the wine during a tasting, might instill in customers.

Charters et al. [24] engaged mystery shoppers to evaluate CDEs in Australia and New Zealand. They found the hospitality of the staff and their ability to create a genuine connection to the winery to be of utmost importance for a positive evaluation of the experience. Additionally, the quality of the experience provided by the staff was more important than the wine quality, which supports the previous findings of O’Neill et al. [25], who investigated the relationship between the perception of service quality and future purchase, re-visitation, and recommendation intentions. Therefore, customers are motivated to visit familiar wineries to experience wine at its source, but also arrive with some apprehension of risks associated with their perception of the wine and the perception of their performance (e.g., the risk of appearing foolish in front of staff and others in their group, and in the CD). However, even though wine can be tasted, thus, removing one risk, it would seem that wine quality has less influence over purchases than the enjoyment of the experience in the CD. As wine is an information-dense subject, information seeking is a risk-reduction strategy, and numerous studies have found education during the experience to be a factor influencing enjoyment of the experience [23,24,26,27].

#### 1.1.3. Cellar Door as an Educational Experience

Education was incorporated into the experience economy by Pine, Gilmore [28], with their new understanding of experiences, which were to be enjoyed as opposed to the selling of goods and services. Duarte Alonso et al. [23] suggested education as a way of reducing the intimidation customers feel, suggesting activities such as stomping on grapes, tastings, and food and wine pairings. Re-visitation intentions were reduced if the opportunity to increase self-efficacy through educational experiences was missing. In their study of the CDE, Ali-Knight and Charters [26] found approximately one third of customers use their CDE to increase their knowledge. While wine-involved customers wanted to know the intricacies of the growing season and vintage, less involved customers wanted to understand their immediate interaction with the wine, why they could or could not taste what was described by the staff member or written in the tasting notes. However, management made assumptions that the educational expectations of customers consisted of technical production information [26]. Furthermore, the provision of professional development (PD) to CD staff to enable them to educate customers consisted only of whatever could be done with minimal investment of time and money. Much of the reviewed research has searched outside the CDE for the magical value-added education or has found that wineries try to incorporate the wrong information into the CDE. In-depth exploration of the CDE is needed to understand what customers want and what staff currently provide. Once the CDE is fully understood, elements contributing to perceived risk can be identified and re-worked into a positive experience for staff and customer.

#### 1.1.4. Cellar Door Staffing

Recently, Lee et al. [29] examined the key attributes affecting visitor satisfaction in a CDE and subsequent spending behaviour. However, their study was based on 480 survey responses from visitors to only one winery. The role of staff was consistently highlighted as being important to various measures of customer satisfaction including, “staff hospitality, which is a key variable in the previous model of satisfaction, remained positive and significant in the spending model” [29] p. 256 and “winery operators should develop strategies to enhance the quality of wine tastings for visitors” [29] p. 256. Research has consistently found similar results, using the same methods; however, none incorporated the staff perspective by exploring the CDE from both sides of the bar. 

Alonso and Kok [30] conducted an in-depth qualitative study of the traits, skills, and personality characteristics future wine industry professionals would need to deliver exceptional CDEs. Drawing their data from interviews with winery owners and managerial staff, they used self-determination theory to elicit five perceived key attributes: attitude, knowledge, strategy, experience, and adaptability. However, their study lacked the input of cellar door staff, who are the employees actually engaging with customers and co-creating the CDE.

An important observation, when reviewing the literature, is that there is no common title for staff who work in a cellar door. They are not winemakers or oenologists, vintners, or sommeliers. Winery owners and winemakers are sometimes found working on a cellar door, but the staff they employ, when they are busy in other areas of the business, do not have a common title, other than staff or assistant. These staff are often casual employees working in a stop-gap role during university studies or friends of the family working for extra cash on a weekend [18]. Larger wineries may employ a cellar door manager, who will often be involved in co-creating CDEs, but the staff who continually create these experiences, found to be so important to the winery business, labour without the respect of an industry-recognized job title or description, security of a permanent position, or defined career pathway. Therefore, an in-depth understanding of the role of staff and their contribution to the co-created experience is missing in the literature. 

#### 1.1.5. Professional Development

In discussions with winery managerial staff, when exploring the nature of Australian wine tourism, d’Ament et al. [21] found the availability of suitable PD models and cost to be the main barriers to implementing the much-called-for PD, especially when staff understandably treat cellar door jobs as stop-gap due to the casual nature of most employment contracts. The Wine Industry Award 2020 does set out a progression of skills (one of which is tour guide) through four grades; however, the second point of each level states, “A person deemed by the employer to have the necessary skills and competence to satisfactorily perform duties at this level” [31]. The award alludes to staff undertaking PD, but then leaves it entirely to the employer to determine the suitability of PD and expertise to the level of employment and pay grade.

Current accredited PD possibilities in Australia include the following:

Certificate III in Wine Industry Operations—contains a specialist unit developed for the cellar door sales stream of the wine sector. The unit provides the skills and knowledge to provide tourist information to winery visitors. 

Wine and Spirit Education Trust (WSET) develops wine tasting skills to an expert level over four courses. Successful students become proficient in evaluative wine tasting and are recognized as authoritative wine specialists.

WSET courses are conducted through private businesses with the completion of all levels at a prohibitive cost, when travel and accommodation are included. WSET providers are based in major cities, with many Australian cellar doors at least a few hours’ drive from major centres. As most cellar door staff are stop-gap or friends of the family, such a personal investment in education without a defined career path is not considered feasible or beneficial. The certificate III was developed for cellar hands and reflects an assumption of the CDE being a tourism activity.

The lack of industry-wide structured PD and job descriptions for CD staff suggests the current research literature is not widely read in the industry, or there is a lack of understanding in how to implement findings. To know that service quality is important is only the first step; understanding *how* the service quality is important requires the exploration of *how* the experience is co-created.

#### 1.1.6. Application of Cognitive Dissonance Theory

It has been established in the literature that customers are employing risk-reduction strategies and staff are working without a clearly defined role, in an industry that is information dense. Therefore, an amount of tension would be expected when beginning a CDE. Cognitive dissonance theory has been used to understand these feelings of unease and tension in professional relationships [32], and various decision-making scenarios [33,34]. Leon Festinger introduced cognitive dissonance theory in 1957, where it quickly gained traction to aid in the understanding of human communication and social influence, and has been widely used in marketing, customer psychology, and social science research (see Harmon-Jones and Mills [35] for an overview). Festinger adopted dissonance to describe the psychological discomfort that resulted from the difference between cognitions. The strength of these feelings of dissonance, between a single cognitive element and established cognitions, depends on both the attribution of importance and number of cognitions, which are consonant or disconsonant with the catalyst of these cognitions (i.e., expectations from a word-of-mouth recommendation of a CDE compared to actual experience). Where individuals choose between contrasting attitudes and behaviours, such contrasts can be reduced by minimizing the significance, or finding new attitudes or behaviours [35]. For example, consider the following possible scenario depicting dissonance: A customer may travel to the CD of their favourite wine, expecting to enjoy their CDE on the site where it was produced. After entering the impressive CD with eager anticipation, they are ignored by staff. After waiting at the counter, they are given only a couple of wines to taste, with no attempt made to engage them in conversation. As it is their favourite wine, the customer buys a bottle of the wine, but leaves having had an unsatisfying experience. The experience leaves them with cognitive dissonance, as they just had an unenjoyable experience with a wine they enjoy. If the wine is very good, they may minimize the importance of the interaction to alleviate the dissonance. However, they may take this new expectation to their next CDE at a different winery. If the service provided at their next CDE is also disinterested and disconnected, then that experience becomes consonant with previous experiences, and the customer may never return to a CD. However, if the service they receive at the second CD is welcoming, engaging, and informative, this may increase the cognitive dissonance toward their previously favourite wine, possibly spoiling their enjoyment of that wine in favour of a newly discovered favourite. The uneasy feelings related to cognitive dissonance have been shown to elicit schadenfreude and trash talking (i.e., negative comments about the brand) to others after a negative experience with a brand [36] in order to restore balance to the emotional state. As Johnson and Bruwer [22] established, customers already approach wine with a perception of risk; therefore, if a negative experience is added, then there is an immediate dissonance with an equally expected hedonic pleasurable activity. 

In this study, cognitive dissonance theory is used to understand how staff co-creating an enjoyable experience could have greater influence over total sales than wine quality. Furthermore, through our exploration of the data gathered from *both* sides of the CDE, a clear framework is provided on which to base PD and staffing models, thus, providing management with a guide not only on how to deliver excellent CDEs, but also reinforcing why enjoyable experiences are so important for retail wine sales.

## 2. Materials and Methods

### 2.1. Research Method, Sample, and Data-Collection

Thirty-three in-depth interviews were conducted in seven different wine regions on the east coast of Australia. The interview duration ranged from 58 min to 97 min. Interviews were audio-recorded with the consent of the participants, who all participated voluntarily and were over the age of 18 (the legal age for consuming alcohol in Australia). A snowballing referral sampling technique was chosen in order to access hard-to-reach and relevant participants, using natural social network processes to help explore unique perspectives [37,38]. Table A1 contains details of the sample. As interviews were conducted during the period from October 2020 to July 2021, coinciding with restrictions due to the COVID-19 pandemic in Australia, this, we acknowledge, may have affected the participants’ recollections and attitudes toward CDEs.

Participants were encouraged to freely communicate their story and their reality of the topic. The interviewer assured anonymity of all businesses and personnel, allowing fluid and open discussion during all the interviews. The interviewer demonstrated a commitment to an authentic co-produced research process and facilitated an open discussion during all the interviews. 

All three researchers have extensive experience in the qualitative study discipline and a strong commitment to communication within wine and tourism research.

### 2.2. Data Analysis

All interviews were transcribed and de-identified by the lead researcher. Emergent coding (where codes are drawn from the tests) was used to find themes using NVivo software (NVivo 12 version 12.7.0 (3873) QSR International, Hawthorn East, Victoria, Australia).

Data was systematically analyzed, consistent with the approach by Charmaz [39], which included initial coding (line-by-line), focused coding, generating categories, and memo writing (see Figure 1). While endeavouring to maintain a systematic approach during the data analysis process, many of these steps coincided, and were not always in order. The constructivist grounded theory methods by Charmaz were modified for this study to generate relevant themes, which are grounded in the data collected from its participants. The constructivist approach acknowledges the role of the researcher as integral to the research process throughout, and the construction of the resultant concept. Constructivist grounded theory is a contemporary version of the original statement by Glaser and Strauss [40]. The constructivist version fosters probing questions about the data and scrutinizes the researcher and the research process. Unlike other versions of grounded theory, the constructivist version also locates the research process and product in historical, social, and situational conditions.

Once themes were coded, content analysis was used to understand participant’s sentiments and feelings towards CDEs. Content analysis is a widely used qualitative research technique defined as a systematic, replicable technique for compressing many words of text into fewer categories based on explicit rules of coding [41]. The method specifically focuses on words, sentence structure, and symbolism (semiotic) to evaluate participants’ perceptions and motivations [42]. The content analysis coding process used ‘clause’ level coding following coding schemes [43].

Data saturation (that is, when no new themes are emerging from the data) became evident in the tenth customer interview and fifteenth staff interview; the remaining interviews were still conducted and coded to enhance the depth of the information collected.

## 3. Findings

Participant’s recollections and perceptions of Australian CDEs placed the beginning of the journey at the budding of a socially outcast Australian wine industry in the 1960s, before wandering through a burgeoning wine tourism industry, which has emerged over the past three decades. The experiences shared highlight a confusion about how CDEs should be constructed, thereby resulting in a disconnection of customers from winery experiences, and management underestimating the necessity of CD staff in creating memorable experiences. While the impact of the winescape on creating a re-connection cannot be underestimated, communicating the story of the industry with the respect Australian wine and wine producers deserve is essential. Furthermore, respect must follow through to the education of both staff and customers, who co-create enjoyable and profitable experiences resulting in lasting connection to the winery and brand. The five main findings in this study were: creating the cellar door experience; over-reliance on the winescape; mutual respect is vital; defining the staff role; and essential professional development. 

### 3.1. Finding 1: Creating the Cellar Door Experience

The Australian wine industry has invested a great deal of time and energy into creating extraordinary wine, while raising the profile of a previously misunderstood profession both domestically and internationally. The industry lacked social acceptance in the Australian society of the 1950s and 1960s, with the current understanding of wine tourism only relevant since the early 1990s. In the intervening three decades of wine tourism, tourists have evolved from busloads of mobile drunks and wine-curious couples [44,45] travelling long distances to experience a cellar door tasting of the well-known brands at the time [46] with nothing to alleviate the disappointment of poor experiences (e.g., restaurants).

The industry began to recognise the need to provide more than ‘a couple of wines’, resulting in the experiences enjoyed nowadays, where it is readily understood that *“we’re in the entertainment business”* S15. Since the first intrepid wine tourists, wine regions and wineries have been encouraged to create ‘value-add’ activities to improve customers’ experiences in the form of food offerings, tours, gift shops, live music entertainment, and gimmicks to increase customer numbers. While providing entertainment for the non-wine-involved visitor, these offerings have disappointed wine-involved customers: *“massive gift shop … a restaurant … rock concerts … changing staff serving the wine …it felt very commercial”* S17. Studies, such as those by Duarte Alonso et al. [23] emphasise the need for these gimmicks to reduce the intimidation created by extremely knowledgeable staff. However, as d’Ament et al. [21] found, without a national wine tourism body, the development of these add-on, value-add, bookable, and commissionable experiences have not been developed with an understanding of the difference between tourism and wine tourism. Winery owners have found it difficult to access clear guidance from the major government associations, Wine Australia and Tourism Australia, who collectively do not understand wine tourism. They have advocated inbound tourism and encouraged wineries to develop experiences for the broad gamut of wine tourist. Such expectations have resulted in a range of experiences being developed to engage all possible customers, from the highly profitable wine-involved repeat visitor to the disinterested ‘hanger-on’, via the inbound tourists who do not join any loyalty programs and avoid the purchase of wine, because they cannot take it home. However, it is widely felt that education delivered by industry associations does not provide the support or education management needed to develop engaging and successful CDEs, delivering *“tourism webinars about inbound, and bookable product that’s commissionable”* S31.

Wineries have embraced expanding their offerings, resulting in multiple business models within the one brand, on the one site. As time and money are invested in research, resulting in premium wine production, being expected to provide non-wine related products has caused considerable consternation and a desire to focus on promoting wine. Customers too can feel overwhelmed by the multiple add-ons offered and would prefer the CDE to be built around the winery’s product: *“would just be nice if it became all about the wine again”* C7. Customers are immediately put at ease if the cellar door is obviously all about wine and wine tasting, as they show little interest in garden tours, gift shops, or restaurants. 

Although Charters et al. [24] and O’Neill et al. [25] found wine quality has limited influence on purchase intentions, this does not necessitate providing alternatives to wine. Australian cellar door customers have evolved in-step with the improving quality and accessibility of Australian wines: *“the younger demographic finds wine fascinating and they want to learn about wine…a unique wine… that wasn’t made in a commercial winery”* S29. Customers’ fascination for wine provides the opportunity for staff to connect with them through the story of the winery and region: *“it was smaller we met with the owner who gave us the full run down… super interesting and I came away with a product that I wouldn’t normally drink because of that experience”* C8. It is the connection between staff and customers that encourage purchases [27].

### 3.2. Finding 2: Over-Reliance on the Winescape

Winescape is a term developed in the tourism literature and is used to refer to the scenic, atmospheric, and structural elements of wine tourism and cellar door experiences [8,9,10]. Tourism Australia and Wine Australia have championed findings from tourism research that suggests wineries need attractions to stand out in their winescape. A tour through any wine region will provide at least one visually impressive cellar door, as architectural extravagance has been embraced to stand out from the competition. However, an impressive cellar door building can overwhelm customers, increasing dissonance and the associated negative feelings. The interior layout can also become a barrier to co-creating a connection.

Reliance on the largess and grandeur of the winescape does not mitigate inattentive or overworked staff, whose inability to communicate with customers prevents connection with the brand, thereby losing customer loyalty and increasing dissonance. Businesses cannot rely on a view or a building to deliver profitable experiences: a building cannot listen to and engage with a customer on an individual basis, and a view cannot tell the stories of a vintage. However, connections created by staff can ease the dissonance: *“impressive building very slick very modern but I didn’t feel intimidated because when you walked in the people were very warm” C7*. Staff tell the story of the winery business and the vagaries of vintage and convey the excitement for newly released wines being offered.

### 3.3. Finding 3: Mutual Respect Is Vital

The third finding is the importance of respect, from staff to customers, and customers to staff, and management to staff.

Cellar door staff are classified by management as retail and not considered to be in the top tier careers: “*ultimately it’s just a casual retail not paid particularly well*” S24. Given the investment in grand cellar door buildings, and the classification of staff in a casual sales role, it would seem that greater respect is given to the inanimate elements involved in the experience than to the staff. Customers, in turn, can reinforce this perception by treating staff as menial, there to serve rather than educate. Furthermore, non-responsive patrons, who are possibly struck dumb due to the dissonance created and unease felt at being in such grand environs, further impact the ability to co-create a positive, enjoyable experience through shared knowledge.

Staff can similarly make judgements of customers, neglecting those who they perceive as being less likely to spend money or changing their demeanour when they realise these customers are going to buy cases. Wine is an information-dense product with a perception of risk; it is understandable that customers approach cautiously and remain disengaged when the industry make wine difficult to understand, for example, using terms such as “textural phenolics”.

There is an art to convivial co-creation generating a warm connection. If missing, it results in a reduction in on-site and post visit sales, with customers unwilling to support businesses offering bad service: *“your wine was great but there’s no way I was going to put my hand in my pocket”* S22. A convivial co-creation relies on the ability to *engage* with customers, rather than delivering the same lines over and over. Tailoring the experience to the customer’s wine knowledge requires a dyadic, co-created conversation. Furthermore, staff need to be able to maintain the flow of the experience when they are busy, without favouring one group over another: *“she was managing a couple of different groups … but would never neglect … always swing back and run you through the next wine … answer any questions …very friendly and welcoming”* C5.

Purchase, revisit, and recommendation decisions are heavily influenced by the human connection, which is either discovered and developed, or lost, during the experience. For the connection to positively influence these decisions, there must be respect. Respect for the wine at the centre of the experience (not all wines are being tasted with a view to buy, many customers taste wine in order to understand more about the varietal, or winemaker’s approach, such as oaked versus unoaked, or the percentages of new oak); respect for the establishment within which the experience exists (while customers may not respond to grand imposing cellar doors, they do expect cleanliness); and respect for each actor co-creating the experience, both the staff and customer. The elevation of cellar door staffs’ status is essential for such mutual respect. 

### 3.4. Finding 4: Define the Staff Role

The elevation of the status of cellar door staff can only begin with winery owners’ acknowledgment of their staffs’ contribution to the profitability of the winery business *“if you’ve got an engaged person they’re certainly buying more”* S24. As mentioned above, there is no industry-wide title for cellar door staff. Management considers cellar door to be a sales role, with a core function of selling wine by delivering well-learnt spiels of wine facts. However, this study shows that creating a positive experience generates word-of-mouth recommendations and involves the engagement of the customer, often letting them dictate conversation content. Happily, some management understand that all sales at the cellar door have the capacity to generate many more through post-visit purchase, and word-of-mouth recommendations. Again, dissonance occurs between what is understood by some sections in the industry (that creating an enjoyable experience sells wine *and* creates enduring customers), and the prevailing attitude, *“it’s just casual retail”* S24.

However, in this study, customers shared how enjoyable CDEs resulted in sales of bottles and cases of wine due to the positive welcome, the story told, and the attention from a staff member, who, through that engagement, built a connection for them with the winery. Additionally, staff members experienced greater enjoyment and job satisfaction when given the freedom to develop rapport with customers, rather than simply recite tasting notes and follow a hard-sell format: *“find out what they enjoy rather than tell them about wine I want to sell them”* C11. A disconsonant assumption is continually created by management, as staff and customers want an enjoyable experience, but employers are labelling cellar door staff as “sales representatives” and conveying the message that chit chat does not sell wine. However, these findings show that being overly sales focused can lose a future enduring customer, whereas creating an enjoyable experience is much more effective when selling wine: “*I walked away with so much wine purely because they were friendly…gave me a bit of a story…added a personality to the wine*” C5.

### 3.5. Finding 5: Essential Professional Development

Widely recognised by staff as an essential requirement to perform adequately in the cellar door, professional development (PD) continues to be found haphazard or lacking [24]. Management attribute inconsistent PD to external sources (i.e., no industry-wide recognised course) or internal sources (i.e., staff not being interested). However, this study found that cellar door staff are interested and willing to participate in PD, simply because it makes their job so much easier: *“the owner always said, ‘people will get bored’ … let me determine when I’m boring the person, but please arm me with the knowledge” S13*. The frustration of staff can negatively affect the experiences provided, and the constant dissonance created by negative assumptions of management contributes to high staff turnover. High staff turnover is another reason management give as to why PD is not provided; management do not want to invest in PD for staff who move to a competitor or into another industry.

Points of disconnection can be created during the CDE through staff being given questionable facts, or making up answers, when much needed knowledge is not provided to them by winemakers or owners. As one customer stated, *“beautiful wines but telling us things that were completely untrue… we didn’t buy”* S17. When education is lacking, all the hard work to develop a connection can be easily undone, thus, re-introducing dissonance and reducing sales, future purchases, re-visitation, and recommendation.

## 4. Discussion

The findings have answered both guiding research questions. Staff establish a relationship within the CDE through non-sales-focused communication, thereby co-creating a convivial connection. The findings very strongly support staff as being the most important element of the CDE for developing a relationship and transforming CD visitors into customers. As we discuss the findings below, we co-create a framework to inform and guide the creation of future experiences, not only in CDs, but wherever businesses benefit from increased sales and brand loyalty through exceptional experiences.

The first finding of this study emphasised the importance of creating a positive CDE. Previous research has similarly emphasised the importance of providing entertaining experiences. Duarte Alonso et al. [23] found tactile interaction with wine making processes (e.g., stomping grapes) reduced the intimidation created by very knowledgeable staff, who were possibly eager to utilise their expensive training. However, the current study went further, recognising and exploring the dyadic relationship at the centre of the experience, providing more in-depth knowledge about the essence of an ideal CDE centred around wine.

The second finding shows the winescape is not a key element for positive CDEs. The current study does not intend to diminish the importance of the winescape, as many customer surveys have found the winescape to be an important motivator to visit wine regions [11]. Nor does it dismiss the impact of grand architectural masterpieces. However, scarce positive affirmation of grand cellar door buildings contributing to positive CDEs was found, whereas recollections of conversations with welcoming and knowledgeable staff were consistently attributed to creating positive CDEs. While these findings do contradict winescape research, perhaps when participants are forced to rate the importance of the various elements of the CDE, including the winescape, a hierarchy is created. For example, when asked in a survey to consider the winescape as a variable contributing to overall satisfaction of a CDE, participants are forced to rate the importance of winescape to their CDE. However, in the current study, the winescape was not readily recalled post-visit.

Mutual respect, the third finding, is essential for any sales and incorporates the importance of respecting all aspects of the CDE, including staff, winescape, and winery ambience. This finding supports the research of Charters et al. [24] and O’Neill et al. [25], which confirms that welcoming and knowledgeable staff who consistently co-create positive CDEs lead to increased sales, re-visitation, and word-of-mouth recommendation, while establishing a relationship that is essential for developing enduring customers.

Although industry professionals recognise the importance of retaining good staff, the fourth finding shows that there is no industry-wide title for CD staff; the role of CD staff is undefined, and without a title. The Wine Industry Award mentions tour guide, which suggests an ability to engage with tourists and convey information, but also leaves the definition of CD staff duties to individual employers. Without a clear understanding of and respect for the role of CD staff, it is not surprising that PD was found to be both essential and lacking.

Widely recognised by staff as an essential requirement to perform adequately in the cellar door, our fifth finding confirms PD continues to be found lacking [24]. The current study includes interviews with both staff and customers and shows that industry professionals recognise the importance of CD staff, but investment in PD of CD staff continues to be haphazard. Staff are essential to a winery’s profitability; therefore, investment in PD must be elevated. Exploring the CDE from both staff and customer perspectives shows that CD staff are interested and willing to participate in PD. Although some managers are open to investing in PD, they are frustrated by the absence of appropriate industry-wide courses.

Failing to provide positive CDEs continues or strengthens dissonance, which leads to word-of-mouth trash talking, rather than word-of-mouth recommendation [36]. A CDE is a dyadic relationship between two or more actors, and developing a connection is key to co-creating a positive experience. Positive experiences lead to purchases; however, this study shows that the CDE should not be sales focused. Instead, the focus should be on building a connection, developing rapport, and co-creating attachment to the brand. Brand attachment creates customer loyalty, delivering future sales and re-visitation, wine club membership, positive word-of-mouth recommendations, and the growth of the customer base.

### 4.1. Proposed Framework

The following proposed framework (see Figure 2) based upon the categories emerging from the current study provides the required elements for ensuring a successful CDE.

Conversations with 33 participants enabled the building of a framework for the co-created CDE. Each participant brought different levels of wine knowledge, different experiences, and perspectives of the wine industry. Each participant’s memories and perceptions of the CDEs provided a deep understanding of the components of and barriers to an enjoyable experience. This framework provides a clear direction for providing a memorable, positive, and enjoyable experience to be co-created and shared with others.

Connection is at the centre of the framework, primarily because without the connection, there are rarely sales, re-visitation, or recommendation. For the connection to dispel the cognitive dissonance, which may have been created by previous experiences, as well as dispel fear of the current environment, or being overwhelmed, there needs to be respect for the needs and wants of the customer *and* the staff member. Their desires are not complicated, nor difficult to deliver. Staff require the attributes to engage a customer in any topic but must be provided the knowledge to answer wine-related questions. Customers want to be genuinely welcomed and engaged, with staff listening to comments and queries rather than simply reciting wine knowledge. Lee and Madanoglu [29] suggest CDEs should be standardised across the industry to deliver relevant wine knowledge to customers. However, our findings clearly show PD should be standardised across the industry, and that although staff need access to relevant facts about vintage, current wine releases, and the history of the winery to satisfy curious customers, both customers and staff prefer a free-flowing and engaging interaction during their experiences. Co-creating CDEs with these abilities provides unique engagement, making the customer feel valued as an individual and not just one of many. This approach is the key to purchases, re-visitation, and recommendations.

### 4.2. Future Research

Development of industry-standard PD for the cellar door workforce is needed. Future research must include a focus on the relevancy of current courses and development options to winery owners, cellar door managers, and the untitled staff creating experiences in the CD. As CDEs are multi-sensory, quantitative studies focusing on the CDE alone, without other tourism activities included in the data set, could provide greater detail about other influences on purchase, re-visitation, and recommendation intentions.

### 4.3. Strengths and Limitations

Research to date has predominantly focused on either the customer viewpoint or management/owner perspective of the CDE. Except for a few qualitative papers analysing customer data, research design has consisted of the quantitative analysis of customer survey data. A strength of the current research is that it used qualitative methods to gain a richer understanding of the cellar door experience from *both* sides of the tasting table. The study also included novice to highly involved wine customers, whose experiences were analysed alongside CD staff and winery owners, with experiences ranging from a few weeks to over six decades.

A limitation to the study was the uncontrollable limitations of travel restrictions imposed due to the COVID-19 pandemic. Therefore, four interviews were conducted via Zoom (an online meeting platform). Due to inconsistent internet connections in regional Australia, the reduced bandwidth was unable to support video, and some interviews only had an audio component. Therefore, some non-verbal communication and cues to question more deeply may have been missed. However, as interviewing continued beyond the saturation of emerging themes, confirming and adding depth to the findings, the effect of the possible missed cues on data was minimal.

## 5. Conclusions

Using an established methodology to understand a different viewpoint of a consumer experience, by exploring both staff and customer perspectives of their CDEs, this study shows that the quality of a customer’s CDE is a vital indicator of the likelihood of purchases, re-visitation, and recommendation. Understanding the elements, which impact the quality assessment, is as important to a wine business as the CD and provides for direct-to-customer sales on-site, the opportunity to develop a relationship, and continues direct-to-customer sales post-visit, which further adds to CD profitability.

Cellar door staff need to be aware that information delivered during a CDE can cause dissonance by overwhelming the customer with too much technical knowledge or disengaging a customer through a lack of welcome or too little knowledge. However, staff can easily reduce dissonance through engaging the customer with information about the wine being tasted and being open to listening to those co-creating the experience. Not all CD visitors are involved wine customers, nor is it the job of the staff member to turn them into involved wine customers; however, showing interest in and engaging with a customer co-creates a positive memorable experience.

Respectful connection is needed for an enduring relationship to emerge, which requires a continuous flow from the warm welcome, through education, and convivial conversation provided during the tasting, to the conclusion of the experience. The connection between staff and customer requires respect for all elements of the experience, through which the bond between the customer, the wine, and the winery is established. Creating the connection through communication and education during a CDE is a dynamic value-add component. Customers can provide a critique of competitors, or local restaurants, with their stories contributing to the staff member’s knowledge. Staff may or may not need to talk about the wine, as the product allows the engagement of multiple senses and the exploration of the intricacies of the depth of colour, flavours, and textures on the palette.

Each participant in the CDE had a unique perspective, understanding, and, ultimately, their own story of not only what they may have learnt, but how they learnt it. The CDE should therefore be approached by staff as an exploratory experience, rather than a standardized scripted performance. Let the tasting notes follow a script. Staff should be aware of how they, the winery, and the wine become part of their customer’s story, and, more importantly, how that customer will re-tell their story to others.

## Figures and Tables

**Figure 1 foods-11-03112-f001:**
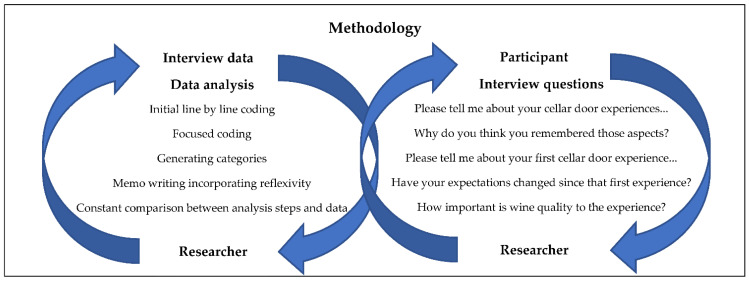
Illustration of how the CGT analysis process is not dependent upon a lineal progression. Through analysis, as the researcher must always re-visit the data, the constant comparison is as integral to the method as constant reflexivity throughout the research. Sample of interview questions also included.

**Figure 2 foods-11-03112-f002:**
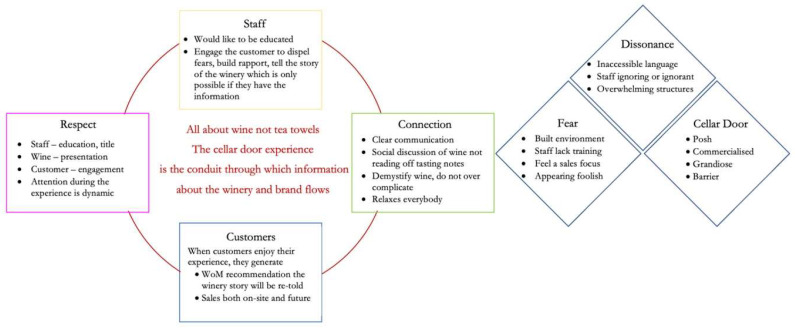
Framework of where to look for and reduce cognitive dissonance and how that may be affecting the ability to deliver positive CDEs.

**Table 1 foods-11-03112-t001:** Perceived risk of wine purchase with risk-reduction strategies in different retail settings.

Risk	Bottle Shop and Cellar Door	
Functional	Wine appreciation and ability to match to foods	
Social	Confidence to select the correct wines, so as not to look foolish in front of family and friends	
Financial	Perceived value for money (not wanting to pay top dollar for unsatisfactory wine, or other wines considered inferior)	
Physical	Effect of wine, such as a hangover or a reaction to wine	
**Risk-reduction strategy**	**Bottle shop**	**Cellar door equivalent**
Information seeking	Knowledge provided by staff	Being on the site of wine production
Brand loyalty	Previous experience with wine	Re-visiting the winery, or previous experience with wine
Store image	Ambience	Style of cellar door (i.e., sleek, or rustic)
Well-known brand	Product readily available in various settings, i.e., restaurants	Readily available and often seen in retail settings, advertised at arts and sporting events, social media, and by word of mouth
Price	Lower bottle price decreases risk	Able to taste, so able to judge value
Reassurance	Connection with staff	Ability to talk to the winemaker, owner, or staff

## Data Availability

Please contact corresponding Author.

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
