# Peer review of "Convivial Connection between Staff and Customer Is Key to Maximising Profitable Experiences: An Australian Cellar Door Perspective"

_foods, 2022, doi:10.3390/foods11193112_

Round 1

Reviewer 1 Report

Dear authors:

The manuscript has discussed the role of several critical factors in co-creating memorable experiences in the cellar door context from both staff and customer perspectives. Based on the in-depth interview and the ground theory, the manuscript proposed a framework for a successful cellar door experience. The manuscript's theme is original, and the paper-writing skill is solid. However, the authors need to consider several issues in the manuscript.

Firstly, I think the existing keywords of the manuscript didn't reflect the core of the study, and maybe there are more appropriate words.

Secondly, I recommend the authors create sub-titles in the literature review section to ensure the description in this section is more ordered and informative. Besides, the content in this section needs more conclusive statements to lead the direction of the literature review to the authors' needs. For now, the content of the literature review is incompact.

Thirdly, in the materials and methods section, I recommend the authors put the ethics approval at the end of the manuscript as it provides personal information. In addition, it would be better for the authors to show more details about the interview outline.

Fourthly, in the findings section, there are two main issues: 1) the authors need to show more about the coding process (i.e., by charts) before describing the five main findings; 2) in most situation, the findings section focus on analyzing data and results, and then discuss the results with prior research in the conclusion and discussion part. Therefore, putting the discussion statements with references to the next section would be much more acceptable.

Fifthly, in the discussion section, when the manuscript concluded different viewpoints compared with prior research, for example, that the winescape was not readily recalled post-visit (page 10, lines 494-497), the authors need to give a more appropriate explanation about it.

At last, it seems there are two different kinds of citation formats that need to be unified according to the request of Foods. For example, ... technical production information (Ali-Knight and Charters, 2001 p 18) (page 4, line 149), d' Ament et al. (2021) found the availability of... (page 5, line 186). And it is also necessary for the authors to ensure the requested formats of the reference.

I'm looking forward to the authors' improving work.

Author Response

The manuscript has discussed the role of several critical factors in co-creating memorable experiences in the cellar door context from both staff and customer perspectives. Based on the in-depth interview and the ground theory, the manuscript proposed a framework for a successful cellar door experience. The manuscript's theme is original, and the paper-writing skill is solid. However, the authors need to consider several issues in the manuscript.

Thank you for your time in conducting a very thorough review of our manuscript.  The authors are very appreciative of your guidance and suggestions which have improved the manuscript adding value to the contribution we’re making to the literature.

Firstly, I think the existing keywords of the manuscript didn't reflect the core of the study, and maybe there are more appropriate words.

Thank you for your comment.  After consultation with peers revised keywords as follows at lines 25-26:

staffing, human resources, cellar door, co-created experience, cellar door experience, brand attachment, constructivist grounded theory, cognitive dissonance, profitability, winescape

Secondly, I recommend the authors create sub-titles in the literature review section to ensure the description in this section is more ordered and informative. Besides, the content in this section needs more conclusive statements to lead the direction of the literature review to the authors' needs. For now, the content of the literature review is incompact.

Thank you for your comment. Sub-titles have been added appropriately throughout the literature review at lines 89, 107, 141, 163, 192 and 222. Concluding sentences have been added to provide a more guided experience for the reader at lines 105-106 and 190-191

Thirdly, in the materials and methods section, I recommend the authors put the ethics approval at the end of the manuscript as it provides personal information. In addition, it would be better for the authors to show more details about the interview outline.

Thank you for your comment. The ethics approval has been relocated at the end of the document. Figure 1 has been added at the end of the methodology section to provide the interview outline, the coding process, and to illustrate the constant reflexivity engaged when following a constructive grounded theory methodology.

Fourthly, in the findings section, there are two main issues:

1) the authors need to show more about the coding process (i.e., by charts) before describing the five main findings;

Thank you for your comment. As previously mentioned in response to an earlier comment the coding process has been included in Figure 1 added at the end of methodology section.

2) in most situation, the findings section focus on analyzing data and results, and then discuss the results with prior research in the conclusion and discussion part. Therefore, putting the discussion statements with references to the next section would be much more acceptable.

Thank you for your comment. Following a CGT methodology requires constant comparison to all data sources, one of which is existing literature.  In this way reporting findings of the analysis can incorporate references to literature to strengthen interpretation and is an accepted reporting style within CGT literature.

Fifthly, in the discussion section, when the manuscript concluded different viewpoints compared with prior research, for example, that the winescape was not readily recalled post-visit (page 10, lines 494-497), the authors need to give a more appropriate explanation about it.

Thank you for your comment. A clearer explanation has been provided at line 504-507.

At last, it seems there are two different kinds of citation formats that need to be unified according to the request of Foods. For example, ... technical production information (Ali-Knight and Charters, 2001 p 18) (page 4, line 149), d' Ament et al. (2021) found the availability of... (page 5, line 186). And it is also necessary for the authors to ensure the requested formats of the reference.

Thank you for your comment. All references have been checked and changed where required to adhere to the requested formats.

Reviewer 2 Report

Dear Authors,

Your work is interesting and appears to be a good contribution to the literature of the wine routes.

However, some minor revisions are recommended!

The methodology appears clear and smooth. Be careful! in line 268 you write table A2 and instead in the appendix there is only tab. A1.

Explain better in the introduction and in the conclusions, why it is assumed that the CDE is a way to increase wine sales, in your opinion.

The Findings in the methodology I would also summarize them in a little mirror in a figure, you can see and quote this work:

Ingrassia, M., Altamore, L., Bacarella, S., Columba, P., & Chironi, S. (2020). The wine influencers: Exploring a new communication model of open innovation for wine producers—A netnographic, factor and AGIL analysis. Journal of Open Innovation: Technology, Market, and Complexity, 6(4), 165.

The bibliography is a bit dated, it should be implemented by including some more recent studies, including other experiences of SDV in Europe.

You can consult and insert this work in introduction and discussions: Ingrassia, M.; Altamore, L.; Bellia, C.; Grasso, G.L.; Silva, P.; Bacarella, S.; Columba, P.; Chironi, S. Visitor’s Motivational Framework and Wine Routes’ Contribution to Sustainable Agriculture and Tourism. Sustainability 2022, 14, 12082. https://doi.org/10.3390/su141912082

Author Response

Your work is interesting and appears to be a good contribution to the literature of the wine routes.

Thank you for your time in conducting a very thorough review of our manuscript.  The authors are very appreciative of your guidance and suggestions which have improved the manuscript adding value to the contribution we’re making to the literature.

However, some minor revisions are recommended!

The methodology appears clear and smooth. Be careful! in line 268 you write table A2 and instead in the appendix there is only tab. A1.

Thank you for your comment, we very much appreciate your attention to detail and have made the appropriate change at line 274.

Explain better in the introduction and in the conclusions, why it is assumed that the CDE is a way to increase wine sales, in your opinion.

Thank you for your comment. Text has been added to the introduction at lines 37-39, and lines 600-603 in the conclusion.

The Findings in the methodology I would also summarize them in a little mirror in a figure, you can see and quote this work:

Ingrassia, M., Altamore, L., Bacarella, S., Columba, P., & Chironi, S. (2020). The wine influencers: Exploring a new communication model of open innovation for wine producers—A netnographic, factor and AGIL analysis. Journal of Open Innovation: Technology, Market, and Complexity6(4), 165.

Thank you for your comment.  A figure has been included at the end of methodology section to provide the interview outline, the coding process, and to illustrate the constant reflexivity engaged when following a constructivist grounded theory methodology.

The bibliography is a bit dated, it should be implemented by including some more recent studies, including other experiences of SDV in Europe.

You can consult and insert this work in introduction and discussions: Ingrassia, M.; Altamore, L.; Bellia, C.; Grasso, G.L.; Silva, P.; Bacarella, S.; Columba, P.; Chironi, S. Visitor’s Motivational Framework and Wine Routes’ Contribution to Sustainable Agriculture and Tourism. Sustainability 202214, 12082. https://doi.org/10.3390/su141912082

Thank you for your comment, one essential component of constructivist grounded theory is to ensure the literature review covers the knowledge developed in the area extending beyond the previous 10 years (Morse et al, 2021. The challenges to and future(s) of grounded theory. Developing grounded theory 2nd edition pg314).  We therefore include papers from the very first explorations of the cellar door experience such as Ali-Knight and Charters (2001) through to Lee, K et al (2021).  The authors also suggest the title of the paper is changed to reflect that this research was conducted in Australian cellar doors.  New Title: Convivial connection between staff and customer is key to maximising profitable experiences: An Australian cellar door perspective.